# Gafchromic™ EBT3 Film Measurements of Dose Enhancement Effects by Metallic Nanoparticles for $^{192}$Ir Brachytherapy, Proton, Photon and Electron Radiotherapy

**Noor Nabilah Talik Sisin** [1], **Raizulnasuha Ab Rashid** [1], **Reduan Abdullah** [1,2], **Khairunisak Abdul Razak** [3], **Moshi Geso** [4], **Hiroaki Akasaka** [5], **Ryohei Sasaki** [5], **Takahiro Tominaga** [6], **Hayato Miura** [7], **Masashi Nishi** [7] and **Wan Nordiana Rahman** [1,*]

[1] School of Health Sciences, Health Campus, Universiti Sains Malaysia, Kubang Kerian 16150, Malaysia; nabilah.talik@usm.my (N.N.T.S.); nasuha_phd@student.usm.my (R.A.R.); reduan@usm.my (R.A.)

[2] Department of Nuclear Medicine, Radiotherapy & Oncology, Hospital Universiti Sains Malaysia, Health Campus, Universiti Sains Malaysia, Kubang Kerian 16150, Malaysia

[3] School of Materials and Mineral Resources Engineering, Universiti Sains Malaysia, Nibong Tebal 14300, Malaysia; khairunisak@usm.my

[4] School of Health and Biomedical Sciences, RMIT University, Bundoora 3083, Australia; moshi.geso@rmit.edu.au

[5] Graduate School of Medicine, Kobe University, Kobe 650-0017, Japan; akasaka@harbor.kobe-u.ac.jp (H.A.); rsasaki@med.kobe-u.ac.jp (R.S.)

[6] Faculty of Health Sciences, Hiroshima International University, Hiroshima 739-2695, Japan; t-tomi@hirokoku-u.ac.jp

[7] Department of Radiation Technology, Hyogo Ion Beam Medical Centre (HIBMC), Tatsuno 679-5165, Japan; h.miura@ham-progress.jp (H.M.); m.nishi@ham-progress.jp (M.N.)

[*] Correspondence: wandiana@usm.my

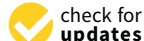



**Simple Summary:** Nowadays, various methods are being investigated to treat cancer patients to reduce radiation to normal organs. The combination of nanoparticles and radiotherapy is one of the active research areas. The presence of nanoparticles could absorb radiation and focus it more on the cancer site compared to the normal cells. This research validated, using dosimetry, that radiation doses could be physically enhanced when combined with higher concentrations of nanoparticles. Thus, this research is fundamental to nanoparticle radiosensitization for different radiotherapy beams.

**Abstract:** Interest in combining metallic nanoparticles, such as iron (SPIONs), gold (AuNPs) and bismuth oxide (BiONPs), with radiotherapy has increased due to the promising therapeutic advantages. While the underlying physical mechanisms of NP-enhanced radiotherapy have been extensively explored, only a few research works were motivated to quantify its contribution in an experimental dosimetry setting. This work aims to explore the feasibility of radiochromic films to measure the physical dose enhancement (DE) caused by the release of secondary electrons and photons during NP–radiotherapy interactions. A 10 mM each of SPIONs, AuNPs or BiONPs was loaded into zipper bags packed with GAFCHROMIC™ EBT3 films. The samples were exposed to a single radiation dose of 4.0 Gy with clinically relevant beams. Scanning was conducted using a flatbed scanner in red-component analysis for optimum sensitivity. Experimental dose enhancement factors (DEF$_{Experimental}$) were then calculated using the ratio of absorbed doses (with/without NPs) converted from the films' calibration curves. DEF$_{Experimental}$ for all NPs showed no significant physical DE beyond the uncertainty limits ($p > 0.05$). These results suggest that SPIONs, AuNPs and BiONPs might potentially enhance the dose in these clinical beams. However, changes in NPs concentration, as well as dosimeter sensitivity, are important to produce observable impact.

**Keywords:** nanoparticles; radiotherapy; dosimetry; clinical beams

## 1. Introduction

The fundamental goal of radiotherapy is to deliver deadly doses to the tumor while avoiding the unnecessary dose toward the surrounding healthy tissues. This conundrum is particularly challenging, especially for percutaneous therapy [1,2]. There are also issues of metastatic cancer treatment difficulties, second primary cancers induced by radiotherapy, effects of cancer and microenvironment interactions, and organ motion during radiotherapy, which might cause treatment complications [2–4].

Currently, there are many ideas on using a combination of radiotherapy with novel chemotherapy as the key strategy to curb the major radiotherapy dilemma [5]. In chemoradiotherapy, researchers formulated ways to improve drug delivery and intensify radiosensitivity at cancer sites [5]. In recent decades, ideas of nanoparticles (NPs) application in radiotherapy, as well as chemotherapy, have been expanding. One of the best received experiments is using metallic nanoparticles (NPs) as dose enhancers [6–8]. Although there are NPs synthesized as natural polymer or lipid NPs, metallic nanoparticles are preferred, as they have high atomic number (Z), which could enhance the radiation-induced cell death [9,10].

Commonly referred to as nanoparticle-enhanced radiotherapy, the direct benefit of this treatment technique is the ability to increase the probability of physical-related radiation interactions. According to Guo, physical dose enhancement (DE) is the direct and linear response from increased radiation interaction with NPs [11]. The photoelectric effect is the dominant interaction process during low-energy photon irradiations. Compton interaction dominates at the photon energy of >0.5 MeV, while pair-production starts at an energy of 1.022 MeV [12]. As a result of photo absorption at the K- and L-shells, the high-Z metallic NPs release energy in the form of secondary electrons (photoelectrons, Auger electrons) and secondary photons (fluorescence) at a highly localized manner to their immediate surroundings, thus creating DE. In a biological scenario, the local DE by high-Z NPs within the tumor creates a contrast against the surrounding low-Z tissue, which would then reduce normal tissue complications while simultaneously increasing the tumor control probability [13,14].

Among the widely proposed metallic NPs in medical research are gold, silver, platinum, iron oxide and bismuth oxide NPs [15–21]. To date, gold (Au; Z = 79) has remained the most studied element for DE because of its high Z and exceptional characteristic [22,23]. Bismuth, having a high-Z number (Bi; Z = 83), is said to be more admirable than Au due to its high X-ray absorption coefficient and affordability [24–26].

Metallic NPs have been investigated for their radiosensitizing properties on cancer cells. Many research works on NPs in preclinical phases are up and coming. It was found that gold NPs (AuNPs) produced different radiosensitization effects in bovine aortic endothelial cells at different X-rays energies (30 to 100 keV) [16]. The presence of AuNPs had also successfully amplified the radiation effects of brachytherapy with the iridium-192 ($^{192}$Ir) source and clinical megavoltage photon and electron in HeLa cervical cancer cells [27]. In addition, different sizes of dendritic platinum NPs (PtNPs) have shown a sensitization factor of 1.77 to 2.31 in HeLa cells under 6 MV photon beam [28]; meanwhile, a factor of 1.10 to 1.23 was indicated in HCT-116 colon carcinoma cells under 150 MeV proton beam [18]. In the past few years, new metallic NPs of bismuth oxide NPs (BiONPs) were suggested as radiosensitizers [29], and there have been several other follow-up research works in 9L gliosarcoma [29] MCF-7 breast cancer [20] and HeLa cells [30] under clinical radiotherapy beams. More importantly, bismuth, as the core of BiONPs, has the advantage of being biodegradable due to its long-standing history in medicine. Its NP form could be readily oxidized and dissolved at physiological conditions and discharged from the human body as soluble bismuth ions [31,32]. Recently, BiONPs effects have been compared in MCF-7 and MDA-MB-231 breast cancer cells and NIH/3T3 normal cells [21,25,33]. The influences of BiONPs when combined with conventional drugs, such as cisplatin, as well as a natural compound, such as baicalein-rich fraction, had revealed that the combinations

could increase radiation doses of clinical radiotherapy beams in cancer cells compared to normal cells [21,25].

Meanwhile, some of the NPs have been investigated in vivo for radiotherapy. Iron oxide NPs in mice exhibited biocompatibility and could be excreted through the liver [34]. Compared with AuNPs and BiONPs, iron oxide NPs have a lower X-ray absorption coefficient due to their lower Z number (Fe; Z = 26), yet are still practical for DE applications [35,36]. When radiation was combined with superparamagnetic iron oxide NPs (SPIONs) in mice, radiosensitization was verified, as tumor growth rate was significantly decreased [37]. The SPIONs could also double as dose enhancers and contrast agents for magnetic resonance imaging (MRI). In the advent of MRI-guided radiotherapy accelerators, the use of multifunctional iron NPs, such as SPIONs, would become very useful in managing cancer patients undergoing radiotherapy [38].

Besides, AuNPs have been demonstrated to have antioxidant properties [39]. A novel protein sulfonic acid reactive AuNPs was reported to improve CT imaging sensitivity and X-rays radiation effects due to the AuNPs uptake and prolonged retention in the living mice [40]. Previous in silico studies of AuNPs in radiotherapy using Monte Carlo simulations had also confirmed the dose enhancement in photon and electron beams [41] and brachytherapy with various sources [42,43]. Compared to other simulated NPs, such as platinum, silver, iodine and iron oxide, AuNPs showed the highest dose enhancement for skin therapy with radiation [44]. Simulations using 3D radiochromic dosimeter and Monte Carlo also discovered the potential of BiONPs' innovative radiosensitizer in clinical radiotherapy applications [45,46].

The radiosensitization process affects the tumor's cellular function by damaging the deoxyribonucleic acid (DNA) directly through single- and double-strand breaks or indirectly via free radical production, such as reactive oxygen species (ROS) through radiolysis. The irradiated tumor cells would then succumb to these damages through many cell death pathways (mitotic catastrophe, apoptosis) after treatment [47]. To further enhance these cell deaths and overcome the limitation of intrinsic tumor radio resistance, researchers have been trying to incorporate radiobiological interventions as a combination to radiotherapy [48].

Although more easily compared to the chemical (i.e., ROS generation, polymerization) and biological (i.e., protein repair blocking) concepts of DE, it is often tricky to measure physical DE strictly. The reason is that most of the measurement techniques are indirect and require the assistance of chemical or biological processes. For instance, the gel dosimetry system responds toward OH• radicals generated during radiolysis [49]. Apart from being expensive and time consuming, using gels means that the measured dose enhancement factor (DEF) may be contaminated with chemical enhancement signals from polymerization reactions with catalytically active NPs [11]. In this study, we proposed radiochromic films (RCFs) as an alternate option for detecting NP-originated physical DE (GAFCHROMIC™ EBT3; Ashland Specialty Ingredients, Bridgewater, NJ, USA). RCF belongs to a class of two-dimensional chemical dosimeters and is well known for its high spatial resolution and near water equivalence [22–24]. Despite operating through the polymerisation of active layer (lithium salt of pentacosa-10, 12-diynoic acid; LiPCDA), the active layer is sandwiched and protected by two chemically inert surfaces of matte polyester substrates. This feature allows for the isolation of physical DE signals from chemical enhancement. Thus, the possibility of additional polymerization of RCFs due to secondary electrons or photons released from NPs can be speculated to be contributed solely by physical DE effects. This approach also has the clear advantage of rapid dose acquisition (minimum 24 h post-irradiation) and the capability of providing direct visualization of enhanced energy deposition [11,50].

The purpose of this study is to investigate the quantification of physical DE using Gafchromic™ EBT3 RCFs with three different types of NPs: AuNPs, SPIONs and BiONPs, as to compare the effects of NPs with different atomic numbers. The present study is the first experimental study that uses different clinical radiotherapy beams to compare different types of NPs. The dosimetry characterization was carried out using standard

film calibration protocols for radiotherapy beams of clinically relevant energies. Precise calibration function and accurate dose measurements are set for the RCF dosimetry system using comprehensive uncertainty analysis. DEF values collected from our experimental setup are then compared against analytical-theoretical DEF calculations from the method outlined by Corde et al. [51] and Roeske et al. [35] for a similar NP concentration and beam spectrum. This study also aims to provide, for the first time, a head-to-head comparison of physical DE quantification between different NPs in a film dosimetry system.

## 2. Materials and Methods

### 2.1. Nanoparticles Synthesis and Characterization: SPIONs, AuNPs, BiONPs

The types, characteristics and properties of all high-Z nanoparticles employed in this study are summarized in Table 1. AuNPs were prepared by diluting the nanoparticles in Dulbecco's phosphate buffered saline (D-PBS) (Gibco, Life Technologies, Carlsbad, CA, USA). Hydrophilic polysulphonic membrane syringe bacterial filter (Sartorius, Gottingen, Germany) of 0.22 μm was used to filter the AuNPs after centrifuging at 1500 rpm for 5 min. The final concentration of AuNPs was obtained by diluting the AuNPs with D-PBS. The SPIONs (10 nm) and $Bi_2O_3$ nanoparticles (60 nm) were prepared by dispersing the nanoparticles with a sterile 0.22 μm hydrophilic polysulphonic membrane syringe bacterial filter (Sartorius, Goettingen, Germany) and then diluting them with D-PBS. The synthesis and characterization of similar types of AuNPs [52], SPIONs [53,54] and BiONPs [24,55] have been reported in the previous literature.

**Table 1.** Summary of properties for SPIONs, AuNPs and BiONPs.

| Type | Core Element Atomic Number (Z) | Average Diameter (nm) | Morphology | Synthesis/Manufacturing Method |
|---|---|---|---|---|
| Iron oxide, $Fe_2O_3$ (SPIONs) | Fe = 26 | 15 | Spherical | Chemical co-precipitation method [53,54] |
| Gold, Au (AuNPs; Au) | Au = 79 | 15 | Spherical | Commercial (AuroVist™) [52] |
| Bismuth oxide, $Bi_2O_3$ (BiONPs) | Bi = 83 | 60 | Rod | Hydrothermal method [24,55] |

### 2.2. Film Dosimeters

In this study, standard issued Gafchromic™ EBT3 films (Lot # 05011702), with initial sheet dimensions of $20.3 \times 25.4$ cm$^2$, were cut into square fragments of $1.0 \times 1.2$ cm$^2$ before being used for calibration and detection of dose enhancement by photon, electron, HDR $^{192}$Ir brachytherapy and proton beam. The individual elemental composition, density (ρ) and effective atomic number ($Z_{eff}$) for each layer described were listed in Table 2.

**Table 2.** Elemental composition and material characteristics of Gafchromic ™ EBT3 films.

| Layer | Composition by Atom (%) [56] | | | | | ρ (g cm$^{-3}$) | $Z_{eff}$ |
|---|---|---|---|---|---|---|---|
| | H | Li | C | O | Al | | |
| Matte polyester substrate (top and bottom; 125 μm) | 36.4 | 0.0 | 45.5 | 18.2 | 0.0 | 1.35 | 6.24 |
| Active compound of LiPCDA (7.5% moisture content; 28 μm) | 56.8 | 0.6 | 27.6 | 13.3 | 1.6 | 1.20 | 7.50 |

### 2.3. Irradiation Setup

Four clinical radiation beams were investigated in this study. Irradiations of photon, electron and $^{192}$Ir brachytherapy sources were conducted at Hospital USM (Kubang Kerian, Malaysia), while proton irradiations were completed at Hyogo Ion Beam Medical Centre (HIBMC, Hyogo, Japan).

#### 2.3.1. Photon and Electron Beam Radiotherapy

Exposures of high-energy photon and electron beams were delivered using PRIMUS™ Linear Accelerator (Siemens Medical Systems, Concord, CA, USA). Two energy levels were selected for each beam quality (photon: 6 MV, 10 MV; electron: 6 MeV, 12 MeV). The source

to surface distance (SSD) and the field size for both irradiations were set at 100 cm and 10 cm × 10 cm, respectively. To conduct electron irradiations, an applicator of choice (same size as field size) was attached to the accelerator's gantry. The Plastic Water® phantom used for photon and electron irradiation was epoxy resin material (Nuclear Associates Inc., Clare Place, NY, USA). In contrast, the bolus was made from a tissue-equivalent gel encased inside a polyurethane 'skin' (CIVCO® Radiotherapy, Orange City, IA, USA). The treatment unit was calibrated to 1 cGy/monitor unit (MU) with a TN30013 Waterproof Farmer ionization chamber manufactured by PTW-Freiburg (Freiburg, Germany). Calibration procedures were performed in compliance with the IAEA TRS398: 2000 Codes of Practice. The irradiation setup for photon and electron beams is shown in Figure 1.

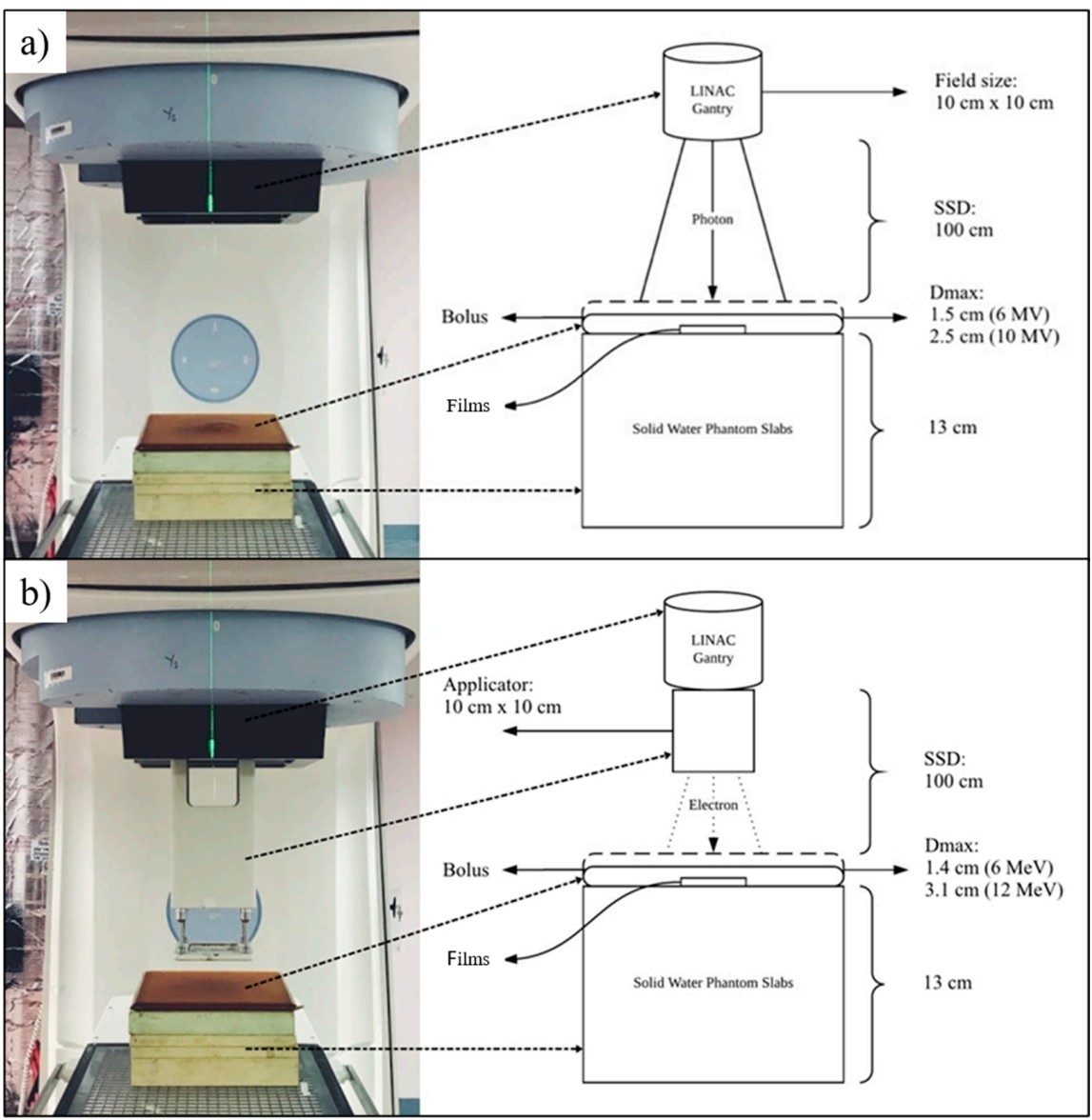

**Figure 1.** The irradiation setup and schematic diagram for (**a**) photon beam and, (**b**) electron beam.

### 2.3.2. $^{192}$Ir HDR Brachytherapy

Irradiations of gamma-ray from Iridium-192 source (0.38 MeV) were achieved using a MicroSelectron® High Dose Rate (HDR) V2 brachytherapy unit (Nucletron, Mallinckrodt Medical B.V., Petten, The Netherlands). The source pellet's length and diameter were measured at 0.06 cm and 0.35 cm, respectively, and it was enclosed inside an AISI 316 stainless-steel capsule. Water-equivalent phantoms and boluses used in this study are

similar to the phantoms from the previous setup. The source was calibrated by employing reference air kerma rate (RAKR) measurement using a well-type ionization chamber of HDR Chamber Type 3304 (PTW-Freiburg, Freiburg, Germany) following the AAPM Task Group 43 protocol [57]. The experimental brachytherapy setup is shown in Figure 2.

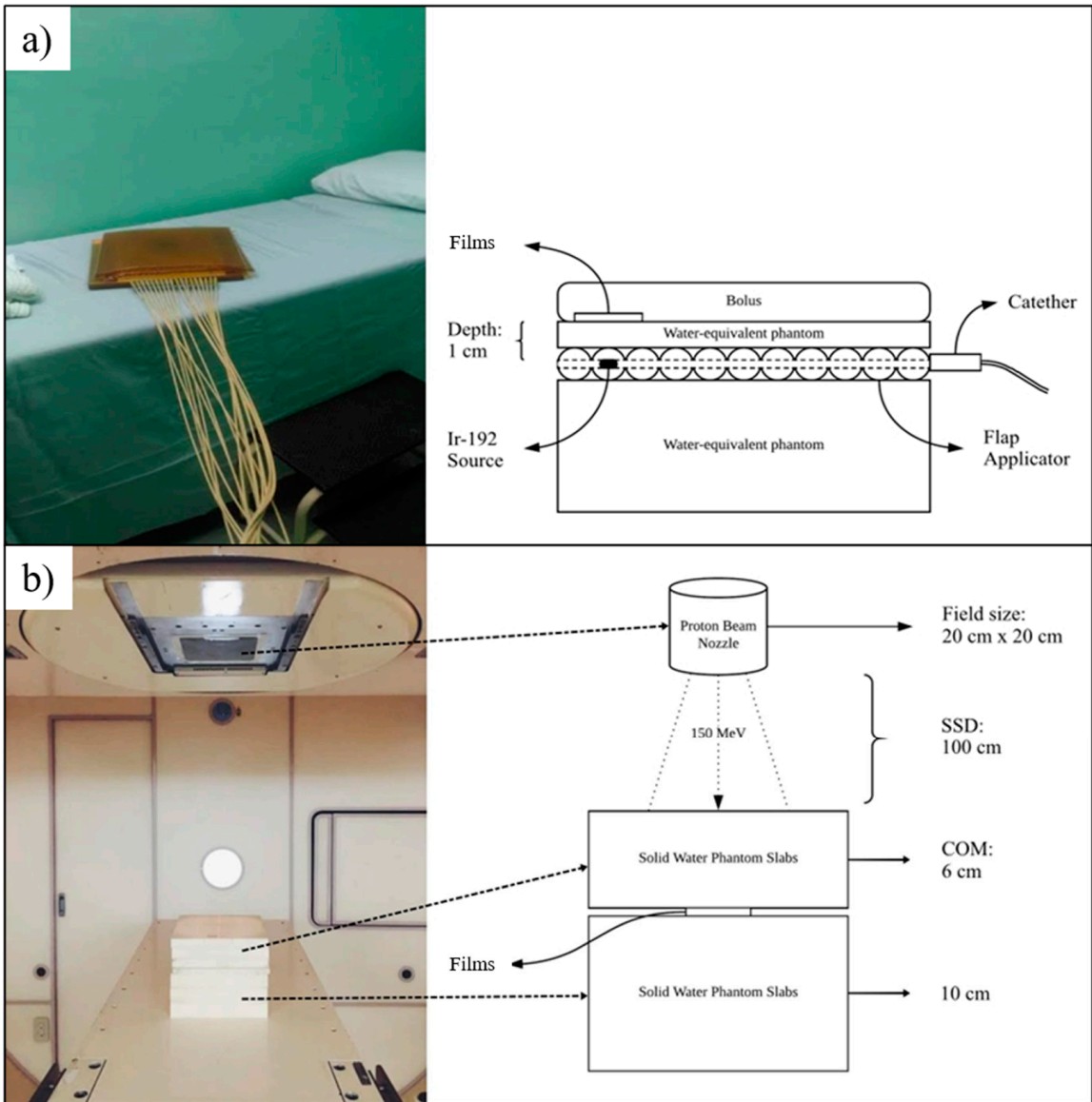

**Figure 2.** The irradiation setup and schematic diagram for (**a**) HDR Brachytherapy and, (**b**) proton beam therapy.

### 2.3.3. Proton Beam Radiotherapy

Proton particle beam accelerator in HIBMC (Mitsubishi Electric, Hyogo, Japan) was operated with an energy level of 150 MeV at a depth of 14 cm. The SSD was set at 100 cm, while the width of the spread-out Bragg peak (SOBP) was within the range of 3 cm to 14 cm (1 cm pitch). Polyethylene-based solid water phantom slabs (Taisei Medical, Osaka, Japan) with dimensions of 20 cm × 20 cm were used throughout the experiments, as shown in Figure 2. Table 3 describes all the irradiation setups.

**Table 3.** Overview description of film irradiation setups.

| Beam Type | Unit | Field Size | SSD | Measurement Depth |
|---|---|---|---|---|
| Photon (6 MV) Photon (10 MV) Electron (6 MeV) Electron (12 MeV) | Siemens PRIMUS™ Linear Accelerator (Siemens Medical Systems, Concord, CA, USA). | $10 \times 10$ cm$^2$ | 100 cm$^2$ | 1.5 cm (Dmax) 2.5 cm (Dmax) 1.4 cm (Dmax) 3.1 cm (Dmax) |
| $^{192}$Ir (0.38 MeV) | Nucletron microSelectron® High Dose Rate (HDR) V2 (Nucletron, Mallinckrodt Medical B.V., Petten, The Netherlands) | $1 \times 1$ cm$^2$ | n/a | 1.0 cm |
| Proton (150 MeV) | Mitsubishi Electric (Mitsubishi Electric, Hyogo, Japan) | $20 \times 20$ cm$^2$ | 100 cm$^2$ | 6.0 cm (COM for SOBP) |

### 2.4. Dose Enhancement Measurement

Clear polyethylene zipper pouches of $1.8 \times 2.0 \times 0.04$ cm$^3$ (KoMee, Ningbo, China) were used in this experiment to contain the media and simulate in-water dose measurements for the Gafchromic™ EBT3 films, as seen in Figure 3. Film calibrations were performed by irradiating the Gafchromic™ EBT3 under full immersion with distilled water. Dose enhancement measurements were conducted with full immersion in 0.5 mL of 10 mM of SPIONs, AuNPs and BiONPs suspended in D-PBS. It is surmised that no significant effect on the mechanical and chemical properties of Gafchromic™ EBT3 films would be observed from using different immersion solutions for the control and experimental groups, as both distilled water and PBS possess only a slight elemental difference, supported by studies on immersion media, such as distilled water and PBS for dental sealer [58,59]. A caveat that should also be considered in this experiment is the penetration rate of liquids into the EBT3 film layers (up to 4.2 mm in 24 h) [60]. Each of the film pieces was immersed for less than 30 min following recommendations by Aldelaijan et al. to reduce any consequential dose error due to the immersion process [61].

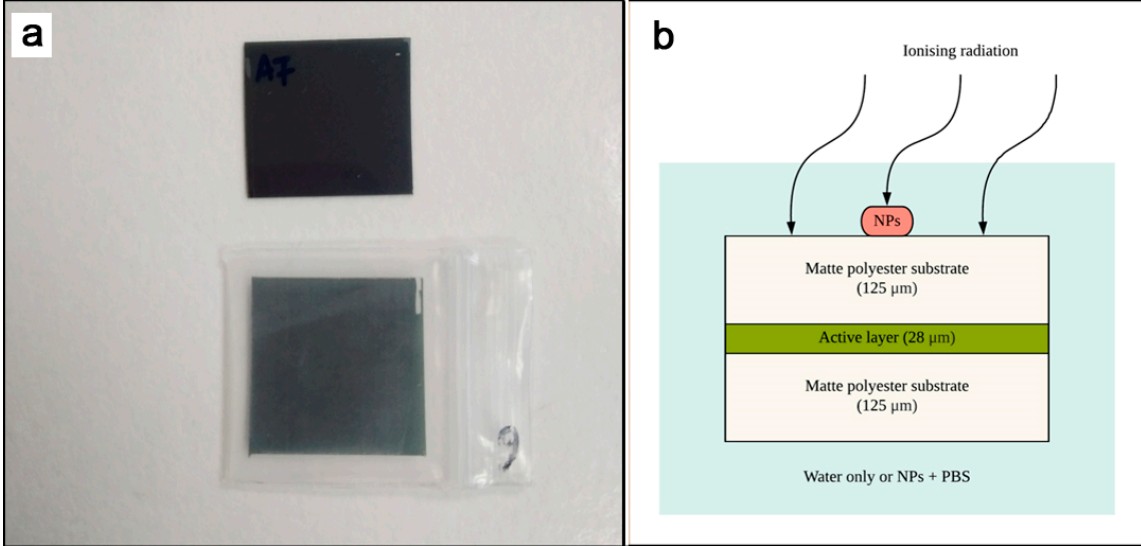

**Figure 3.** (**a**) 1 cm $\times$ 1 cm Gafchromic™ EBT3 films outside and inside the clear polyethylene zipper bag, and (**b**) the cross-sectional illustration of EBT3 films immersed inside the enclosure with either distilled water or NP-loaded saline water.

The films were then patted dry using a soft microfiber cloth and stored inside a light-tight envelope in refrigerated conditions (~4 °C) for 72 h. These measures were taken to ensure the LiPCDA polymerization stabilizes and further decreases the change in optical density incurred from the in-water measurement [61,62]. The experimental process for both

calibration and dose enhancement measurement was repeated independently three times to reduce statistical uncertainty.

### 2.5. Film Calibrations and Analysis

The Gafchromic™ EBT3 films pieces were handled individually using latex gloves and clean wiped using a 70% ethanol-dampened soft microfiber cloth before scanning to reduce systematic errors. An EPSON Expression® 10000XL flatbed scanner (Epson Seiko Corp., Nagano, Japan) was used to acquire film images in 24-bit red green blue (RGB) and 150 dpi digital resolution, consistent with the clinical practice. The films were strategically positioned at the centre of the scanner bed to minimize lateral response uncertainty. After scanning was completed, the transmission pixel value signals from the red component were extracted from the output images of uncompressed tagged image file format (TIFF) using FilmCal software v2.4 (PTW-Freiburg, Freiburg, Germany).

To prepare the images for response analyses, all image adjustments, color and flattening correction options, and lookup tables were disabled or reset before scanning activities. Default scanning parameters of 150 dpi resolution, 24-bit depth RGB and no image inversion were adopted into the PTW FilmScan and EPSON Scan software. The average region or region-of-interest (ROI) feature on PTW FilmCal software version 2.4 (PTW-Freiburg, Freiburg, Germany) was then used to gather raw, red channel *I* information. Red channel was chosen as a default scanning parameter because it holds the highest sensitivity among the three RGB channels and is most suitable to the tested dose interval, which is below 8 Gy [63]. RCF response relied on net optical density (*netOD*), which is associated with *I* values through the Beer–Lambert relationship. Therefore, measurements of *netOD* were carried out using Equation (1):

$$netOD = OD_{exp} - OD_{unexp} = log_{10}\left(\frac{I_{unexp}}{I_{exp}}\right) \tag{1}$$

$OD_{exp}$ refers to the optical density of the exposed film, $OD_{unexp}$ refers to the optical density of the unexposed film, $I_{unexp}$ refers to the averaged pixel value of the reflected intensity in unexposed film pieces, and $I_{exp}$ refers to the averaged pixel value of the reflected intensity in exposed film pieces.

To assess *netOD* value uncertainty or film reproducibility, the standard deviations (SD) of the mean pixel values obtained from the ROIs were calculated using Equation (2):

$$\sigma_{netOD} = \frac{1}{ln(10)} \cdot \sqrt{\left(\frac{\sigma_{I_{unexp}}}{\sigma_{I_{unexp}}}\right)^2 + \left(\frac{\sigma_{I_{exp}}}{\sigma_{I_{exp}}}\right)^2} \tag{2}$$

$\sigma_{I_{unexp}}$ and $\sigma_{I_{exp}}$ refer to the SD of the measured $I_{unexp}$ and $I_{exp}$. Film non-uniformity, on the other hand, was calculated using the standard deviation from the mean in the ROIs of each film piece.

To investigate GAFCHROMIC™ EBT3 films' dynamic range in red channel, a dose–response curve was plotted from the calculated *netOD* against the prescribed dose, *D*. The non-linear *netOD* power-fitting method by Devic et al. [64] was chosen for the construction of said dose–response curve, as indicated in Equation (3):

$$D = B \cdot netOD + C \cdot netOD^n \tag{3}$$

*B* and *C* were the fitting function's parameters. The exponentiation of *netOD*, *n*, was introduced based on RCF's non-linear saturation response at high doses, and it has been identified as an interchangeable parameter to suit different experimentation setups. To determine the best *n* value for the fitting function, a comparison was made between *n* = 2.0 (Equation (4)) and *n* = 3.0 (Equation (5)) to resemble quadratic and cubic polyno-

mial functions, which has also been reported to provide a sufficiently accurate fit in low doses [65,66].

$$D = B \cdot netOD + C \cdot netOD^2 \tag{4}$$

$$D = B \cdot netOD + C \cdot netOD^3 \tag{5}$$

The Nonlinear Curve Fit feature inside OriginPro 2018 software fits the datasets. To further optimize the modified polynomial fitting method, orthogonal distance regression (ODR) algorithm was used instead of the commonly employed Levenberg–Marquardt (LM) quasi-Newton algorithm. This regression method allows uncertainty evaluations for independent variables by minimizing the sum of squared orthogonal distances between each data point and the fitted curve [67]. The goodness-of-fit will then be examined using four sets of tests: visual, the coefficient of determination and adjusted coefficient of determination ($R^2$, adj-$R^2$), reduced chi-square ($\tilde{x}^2$) and Akaike's information criterion (AIC) test.

### 2.6. Experimental Dose Enhancement Analysis

$DEF_{Experimental}$ is defined as the increase in the estimated absorbed dose measured during NP immersion regarding that produced in similar conditions in water-only immersion for the same beam quality. In the context of Gafchromic™ EBT3 films, the $DEF_{Experimental}$ is calculated using Equation (6):

$$DEF_{Experimental} = \frac{D_{fit_{NP}}}{D_{fit_{control}}} \tag{6}$$

whereby, $D_{fit_{NP}}$ is the estimated absorbed dose measured in a medium with NP (PBS + NP; converted from the calibration curve plotted from *netOD* values), and $D_{fit_{control}}$ is the estimated absorbed dose measured in a medium without NP (distilled water). Evaluation of experimental dose enhancement for every NP was performed at a single dose point of 4.0 Gy, which lies within the sensitivity range of the red component for Gafchromic™ EBT3 films. Two-sample *t*-tests were performed using OriginPro 2018's Hypothesis Testing analysis to determine the significance of the quantified DEF ($p < 0.05$ is statistically significant).

### 2.7. Theoretical Dose Enhancement Analysis

To theoretically estimate the DEF caused by NPs for monoenergetic beams, the non-Monte Carlo modeling method of calculating variations of mass energy absorption coefficient in various beam energies (1 keV to 20 MeV) was used [51], stated in Equation (7):

$$DEF = \frac{\left[\frac{\mu_{en}}{\rho}\right]_E^{water+NPs}}{\left[\frac{\mu_{en}}{\rho}\right]_E^{water+NPs}} = \frac{w_{NPs} \cdot \left[\frac{\mu_{en}}{\rho}\right]_E^{NPs} + (1 - W_{NPs}) \cdot \left[\frac{\mu_{en}}{\rho}\right]_E^{water}}{\left[\frac{\mu_{en}}{\rho}\right]_E^{water}} \tag{7}$$

where $\left[\frac{\mu_{en}}{\rho}\right]_E$ is the mass energy absorption coefficient for the target (water, NPs or water + NPs) at a specific monochromatic X-ray beam energy (*E*), and $w_{NPs}$ refers to the weight fraction of NPs in the water. Values of mass energy absorption coefficient were obtained from the United States National Institute of Standard and Technology (NIST) X-ray Mass Attenuation Coefficients database [68].

## 3. Results

### 3.1. Calibration Curve

Figure 4 shows the dose–response curves of standard Gafchromic™ EBT3 films for different fitting functions. From the red channel dose–response graphs, it can be seen that both fitting Equations (4) and (5) had provided an almost identical, visually smooth fit shape. There were no Runge's phenomena or oscillatory tendencies exhibited in the curvature profile of *n* = 3.0, which is commonly seen in higher polynomial order [69]. The

standard Gafchromic™ EBT3 Films' response is nearly independent of all investigated radiotherapy modalities, except for Iridium-192, which produces a slightly higher response across all prescribed doses. Table 4 shows the example of *netOD* values at 250 cGy for all modalities and energies.

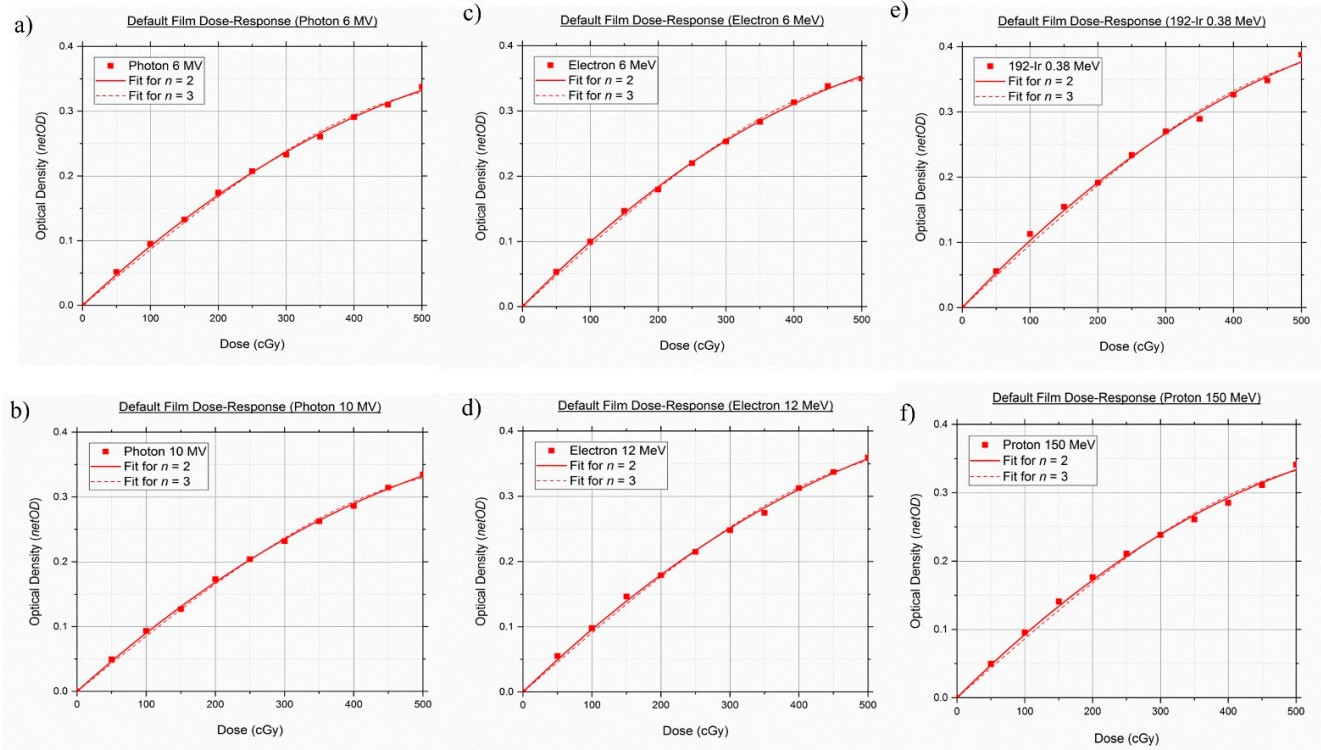

**Figure 4.** Default dose–response curves of standard GAFCHROMIC™ EBT3 films in two fitting functions: (**a**) Photon 6 MV, (**b**) Photon 10 MV, (**c**) Electron 6 MeV, (**d**) Electron 12 MeV, (**e**) Iridium-192 0.38 MeV, (**f**) Proton 150 MeV.

**Table 4.** Values of *netOD* at 250 cGy for standard Gafchromic ™ EBT3 films.

| Film Response (*netOD*) | Photon | | Electron | | 192-Ir | Proton |
|---|---|---|---|---|---|---|
| | **6 MV** | **10 MV** | **6 MeV** | **12 MeV** | **0.38 MeV** | **150 MeV** |
| *D* = 250 cGy | 0.2073 | 0.2040 | 0.2200 | 0.2148 | 0.2342 | 0.2108 |

The goodness-of-fit tests ($R^2$, adj-$R^2$, $\tilde{x}^2$ and AIC) for the standard Gafchromic™ EBT3 films were tabulated in Table 5. Coefficient of determination and its adjusted version for $n = 2.0$ and $n = 3.0$ yielded a value of 1 in all testing, suggesting that both fitting functions fits the datasets appropriately. However, it can be seen from the data in Table 5 that the reduced chi-square value for $n = 2.0$ was lower than the values obtained from $n = 3.0$. A similar trend was observed for the AIC tests, as $n = 2.0$ exhibited lower AIC values in comparison to $n = 3.0$. Taken together, these findings provide a clear indication that the quadratic-like function of Equation (3) provided the best agreement among the four goodness-of-fit tests for all investigated beam types. Hence, Equation (3) was then used for each subsequent curve fitting.

**Table 5.** Fitting analysis in terms of $R^2$, adj-$R^2$, $\tilde{x}^2$ and AIC for standard Gafchromic ™ EBT3 films.

| Beam Type | n | $R^2$ | adj-$R^2$ | $\tilde{x}^2$ | AIC |
|---|---|---|---|---|---|
| Photon (6 MV) | 2.0 | | 1 | 1.50 | −114.98 |
| | 3.0 | | | 4.73 | −102.32 |
| Photon (10 MV) | 2.0 | | 1 | 1.00 | −119.41 |
| | 3.0 | | | 3.11 | −106.96 |
| Electron (6 MeV) | 2.0 | | 1 | 0.89 | −120.74 |
| | 3.0 | | | 2.27 | −110.39 |
| Electron (12 MeV) | 2.0 | | 1 | 1.94 | −112.11 |
| | 3.0 | | | 4.97 | −101.79 |
| $^{192}$Ir (0.38 MeV) | 2.0 | | 1 | 51.42 | 50.56 |
| | 3.0 | | | 106.43 | 58.56 |
| Proton (150 MeV) | 2.0 | | 1 | 31.19 | 45.06 |
| | 3.0 | | | 75.95 | 54.85 |

### 3.2. Theoretical Dose Enhancement by SPIONs, AuNPs, BiONPs

Figure 5 shows the DEF as a function of monoenergetic energies (1 keV to 20 MeV) for different elements at different concentrations in water, from 1% to 100%. As mentioned in the methodology section, the data represented in Figure 5 were interpolated from the NIST database, which features the mass energy absorption coefficient of elements and substances of dosimetric interest. The theoretical DE of the core metallic Fe, Au and Bi represented the theoretical DE of SPIONs, AuNPs and BiONPs, respectively. It is evident from the results that the theoretical DEF computations showed energy, concentration and Z number dependence, consistent with previous calculations on elements such as iodine, gold and bismuth [16,51,70].

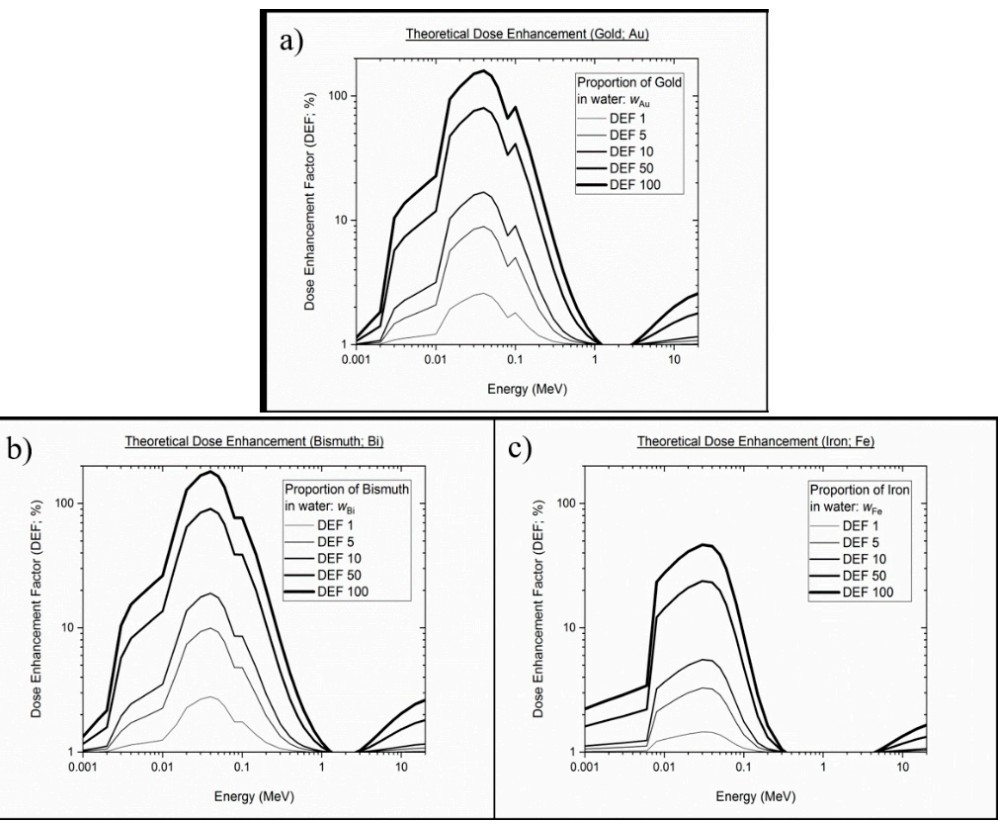

**Figure 5.** Theoretical dose enhancement factor of core of the metallic elements at various energy ranges and concentration percentages in water: (**a**) Gold, (**b**) Bismuth, (**c**) Iron.

### 3.3. Dose Enhancement by SPIONs, AuNPs, BiONPs

Figure 6 and Table 6 show the physical DEF obtained from the measurements using Gafchromic™ EBT3 films for different NPs irradiated with different radiotherapy beams. Experimentally, the NP-immersed EBT3 film for HDR $^{192}$Ir was shown to posses DEF values of higher than 1, with mean enhancement signals ranging between 0.8–3.5%. The level of consistency with respect to the DEF$_{\text{Theoretical}}$ is somewhat similar because it increased with increasing Z number, with BiONPs showing the highest experimental dose enhancement, followed by AuNPs and SPIONs. For 6 MV photons, the experimental DEF matched its theoretical pairing, i.e., no dose enhancement effects were able to be shown for all NPs. Although the analytical calculations for 10 MV photons predicted an enhancement in AuNPs and BiONPs, the Gafchromic™ EBT3 films were unable to detect any increase in dose experimentally.

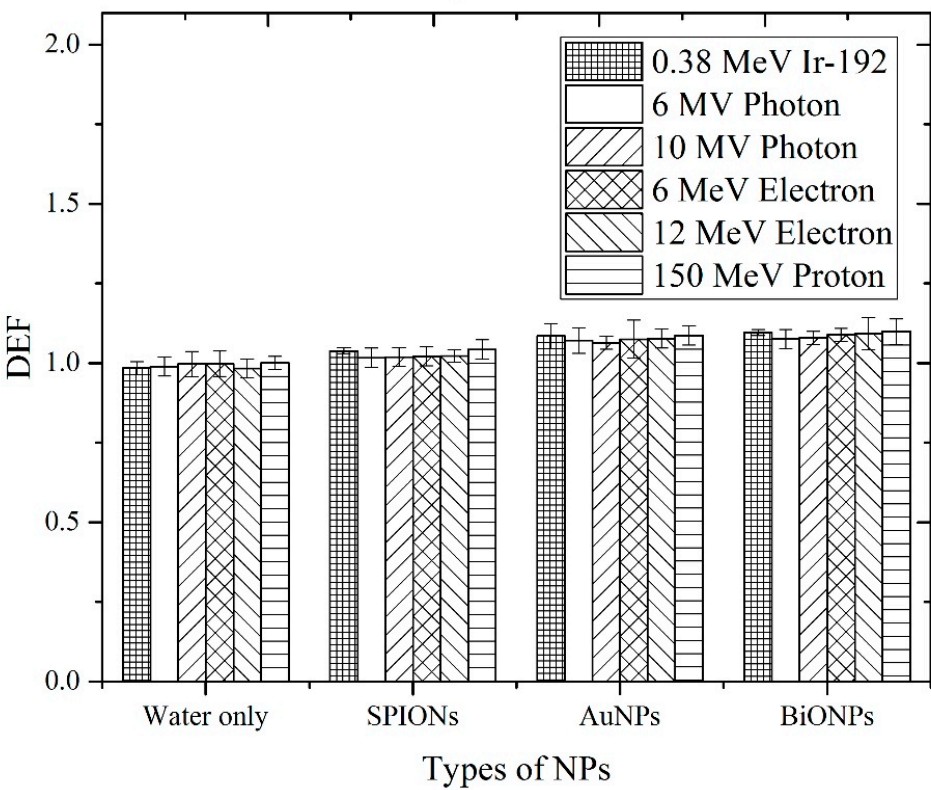

**Figure 6.** The physical DEF of different NPs obtained from the measurements using Gafchromic™ EBT3 films under several types of clinical radiotherapy beams (Ir-192, photon beams, electron beams and proton beam).

**Table 6.** The physical DEF obtained from the measurements using Gafchromic™ EBT3 films.

| Immersion Media | 0.38 MeV $^{192}$Ir | 6 MV Photon | 10 MV Photon | 6 MeV Electron | 12 MeV Electron | 150 MeV Proton |
|---|---|---|---|---|---|---|
| Water only | 0.98 ± 0.02 | 0.99 ± 0.03 | 1.00 ± 0.04 | 1.00 ± 0.04 | 0.98 ± 0.03 | 1.00 ± 0.02 |
| SPIONs | 1.04 ± 0.01 | 1.02 ± 0.03 | 1.02 ± 0.03 | 1.02 ± 0.03 | 1.02 ± 0.02 | 1.04 ± 0.03 |
| AuNPs | 1.09 ± 0.04 | 1.07 ± 0.04 | 1.06 ± 0.02 | 1.08 ± 0.06 | 1.08 ± 0.03 | 1.09 ± 0.03 |
| BiONPs | 1.10 ± 0.01 | 1.08 ± 0.03 | 1.08 ± 0.02 | 1.09 ± 0.02 | 1.09 ± 0.05 | 1.10 ± 0.04 |

Under all of the investigational conditions, no substantial dose enhancement due to the addition of NPs was detected. The two-sample t-tests verified the conclusion put forward, as the *p* values were larger than 0.05 in almost all of the datasets, showing insignificant differences between the water-only and the NP groups. Although HDR $^{192}$Ir + BiONPs

were able to show statistically significant $DEF_{Experimental}$ ($p < 0.05$), the enhancement cannot be appreciated in full due to its modest amount. An experiment using a proton beam to observe the effects of DE by heavy charged particles also showed a slight increase in DEF.

## 4. Discussion

Analytical calculations are one of the many methods used by researchers to predict physical DE by photons. Corde et al., for example, uses Equation (7) to estimate the DEF generated by 10 mg/mL iodine at monoenergetic photon energies of 0.001 MeV to 0.5 MeV [51]. Roeske et al. studied the theoretical DE on multiple NP elements (Z = 25–90; 5 mg/mL) using clinically relevant photons of external beams and radionuclide sources [35]. In a more recent publication, Srinivasan et al. used the mass energy absorption coefficient for soft tissue (ICRU-44) as the low Z interface instead of water to simulate the DEF generated by 7 mg/mL AuNPs [71]. It is evident from the literature that the analytical calculations were conducted under varied conditions, i.e., element type, photon energy and concentration. Although it served as a challenge for us to conduct side-to-side comparisons, the results obtained in our calculations were in line with the trend of DEF observed in the studies mentioned above. In this paper, we first demonstrated that the incorporation of high Z component into the analytical calculation of absorbed dose generates theoretical DE. Bismuth oxide, characterized with the highest Z number among the three elements, offered the greatest $DEF_{Theoretical}$, followed by gold and then iron NPs. This is especially true for HDR $^{192}$Ir because of the larger amount of low-energy photons in its spectrum (0.296–0.608 MeV), where Z plays a major role in dose enhancement. In contrast, the $DEFs_{Theoretical}$ for 6 MV and 10 MV photons were far lower due to the dominance of interaction that is independent or weakly dependent on Z, i.e., Compton interaction.

To the best of our knowledge, this is the first study that calculates analytical $DEF_{Theoretical}$ for different types of elements using Boone and Chavez's total mass energy absorption data source via the XMuDat program. The referred data were chosen for this research because they cover all of the subsequent re-emission and reabsorption possibilities generated when an incidental photon travels through an absorber medium [72]. In opposition to the more commonly used data source by Hubbell and Seltzer in the XCOM program/NIST website, the generated $DEF_{Theoretical}$ may be underestimated because it has been reported to exclude the contribution of secondary photons, i.e., fluorescence [35]. This could explain our findings, whereby the maximum $DEFs_{Theoretical}$ for AuNPs and BiONPs occur at a slightly lower energy (0.038 MeV) compared to those reported by Hubble and Seltzer's (0.040 MeV) [11,12,36]. After recalculation, the $DEFs_{Theoretical}$ for Au and Bi from our study (Au $DEF_{Theoretical}$ = 9.364; Bi $DEF_{Theoretical}$ = 26.251) were also found to be slightly higher than Hubble and Seltzer's (Au $DEF_{Theoretical}$ = 9.244; Bi $DEF_{Theoretical}$ = 24.571), possibly indicating the added contribution of fluorescence. Nonetheless, the analytical approach is not ideal and is still lagging behind more sophisticated methods, such as the computerized Monte Carlo (MC) system. This data-intensive method is capable of simulating many more geometrical aspects of the DE. For example, MC such as GEANT4, MCNP5 and EGSnrc, can be programed to randomly sample the path of radiation-induced products, generate intricate geometries and model more complex interactions (backscattering of secondary electrons, Auger avalanches, succeeding Compton electrons/scattered photons interaction, plasmon excitations) [73]. The analytical calculation here only considers a linear trajectory framework and assumes that all of the energy generated would deposit on the intended target. Such assumptions and simplifications were made to achieve the simplest method of predicting physical DE but may require further investigations with MC systems to cross-validate the findings.

Meanwhile, initial tests were conducted on the film dosimetry system to establish an optimized protocol for accurate calibration and DE measurements. The selected *n* parameter satisfied all of the fit tests and was found to produce an acceptable combined standard uncertainty ($k = 1.0$; <5.00% of radiotherapy accuracy) [74]. Most film dosimetry studies have investigated the ability of high-Z contrast materials or metallic NPs to induce

DE in both monochromatic and polychromatic low-energy photon beams (the majority of which use a single type of NPs) [75–78]. The presence of NP-loaded solution in the dosimeter's environment resulted in an increase in absorbed dose. However, the under-coverage for deep-seated tumors by photons of this spectra limits their potential DE application. Any radiation dose delivered from this poorly penetrating beam would also have detrimental effects to the skin and skull if not properly filtered and tuned for intensity modality radiotherapy [79]. Conversely, the photon modalities used in this study are supported by currently available clinical practice, but they seemed to be less in harmony with the sought-after photoelectric effect.

It was demonstrated in Figure 6 that the $DEF_{Experimental}$ for water only and SPIONs were in fair agreement with $DEF_{Theoretical}$ (from Figure 5) across all photon beams. SPIONs, having a relatively low Z number, were expected not to produce substantial physical DE [80], and most of the radiosensitization effects observed in biological samples during SPIONs-photon irradiation were dominated by the effects of ROS [36,81]. The $DEFs_{Experimental}$ for AuNPs and BiONPs, on the other hand, were found not to be in agreement with the $DEF_{Theoretical}$ (except for 6 MV photon). The $DEF_{Theoretical}$ also showed no significant difference among the different radiotherapeutic techniques within the same NPs. These discrepancies can be attributed to the fact that the $DEF_{Theoretical}$ considers all the energy generated from the NP–photon interaction would be deposited at the intended target, whereas the detection geometry limits the $DEF_{Theoretical}$. In our experimental setting, the target site is the active layer of EBT3 films. Based on the $\sigma_{DEF_{Experimental}}$ and the statistical test, we concluded that no considerable $DEFs_{Experimental}$ were able to be recorded by the active layer, and they are said to occur mainly in three parts:

1.  The abundant, low-energy electrons, i.e., Auger electrons produced by NPs had nanometric trajectory range and were hypothesized to be physically absorbed or attenuated by the outer layer of matte polyester substrate;
2.  Highly energetic electrons, i.e., photoelectrons generated by the NPs at the specific concentration, with trajectory range of >125 µm, were sparse;
3.  The measured $DEFs_{Experimental}$ due to photoelectrons are lower than their associated uncertainty and statistically indistinguishable from the control experiment.

Regardless, the high amounts of ROS during AuNPs and BiONPs irradiation with photons were the primary cause of DE (low cell survival fractions), rather than the physical contributors [1,16,25].

The focus on NP dose enhancers has been attributed mainly to their interaction with photon radiation. As the number of heavy-particle facilities [82] is growing, so are the studies investigating DE by NPs during heavy-particle irradiations. Proton radiotherapy has a unique advantage over conventional photon radiotherapy because of the Bragg peak, which allows for excellent dose conformity to the tumor. In a pioneering study by Kim et al., the group demonstrated that AuNPs and SPIONs were capable of enhancing proton irradiation in vivo, as they observed a 75% to 90% mouse tumor volume reduction, following an intravenous NP injection with 41.7 MeV proton irradiation [83]. Among the contributors suggested for prompting DE during proton irradiations are particle-induced X-ray emission (PIXE), particle-induced gamma-ray emission (PIGE), secondary electrons, nuclear activation products, i.e., ions (neutrons, electrons, protons, deuterons or tritons) and recoil nuclei—on top of chemical–biological responses, such as ROS [83,84].

Until now, there has only been a handful of research attempts to assess the physical DE by NP–proton interaction using film dosimetry [84,85]. The film measurements conducted in this study showed no measurable DE across all types of NPs under 150 MeV proton irradiation—which shares a similar conclusion with Cho et al. [84]. In the same way as photons, protons are also capable of generating an avalanche of secondary electron emission from high-Z NPs through impact ionisation and ensuing Auger cascades [6]. Nevertheless, MC simulations have shown that the secondary electrons produced from proton-irradiated NPs travel in a much shorter range than those generated from photon-irradiated NPs. The use of EBT3 RCFs may provide an adequate detection geometry for

highly energetic PIXE/PIGE but most definitely not suitable for the weaker and shorter-ranged Auger electrons due to the thick polyester substrate. Surprisingly, Ahmad et al. managed to observe a substantial $DEF_{Experimental}$ (1.210 or 21.0% DE for 5.5 mg Au/mL; 1.260 or 26.0% DE for 1.1 mg Au/mL) using EBT3 films under proton irradiation [85]. A plausible explanation for the high $DEF_{Experimental}$ might be due to the different proton beam energy (226 MeV), concentration and NP size (50.7 nm). Further validation studies with the same experimental setup are required to confirm the findings.

The increased cell deaths or tumor volume reduction shown in biological studies during proton irradiation were also mainly due to the enhanced ROS generation [8], with the latest simulations suggesting the most considerable proton-induced ROS enhancement occurring over 50 nm from the NPs' surface [86]. Our in vitro model (with HCT116 human colon cancer cell) confirmed the idea, as the addition of iron-, gold-, bismuth oxide- and platinum-based NPs during proton irradiation was found to spike the amount of ROS [17]. The proposed mechanisms of enhanced ROS generation in cells were attributed primarily to the emission of secondary electrons from NPs. Therefore, to comprehensively and accurately quantify DE effects exclusively from the physical standpoint, a combination of chemistry and physics modeling via MC with biological work is much needed.

## 5. Conclusions

Overall uncertainties in default scanning parameters were kept at a precision requirement of 5% in all beam modalities and energies. In all beam modalities and energies, higher Z elements, higher concentrations and the kilovoltage range were found to have higher theoretical DEF of the three metallic NPs. However, the physical DEFs caused by radiation beam interactions with AuNPs, BiONPs and SPIONs were negative. Despite the absence of considerable physical DEFs, NPs have an impact because biological impacts are mostly reliant on ROS, which NPs produce more of. Beam energy in the kilovoltage range may produce more physical DE or positive outcomes. Due to the thick polyester substrate, the usage of EBT3 RCFs may render an appropriate detection geometry for highly energetic PIXE/PIGE, but it is unsuitable for the weaker and shorter-ranged Auger electrons. Nonetheless, the result is a reasonable analog for prior in silico work on NPs position dependency. In order to establish the best optimized approach that may suit the current material supplies and irradiation access, conventional and non-conventional film dosimetry strategies were investigated.

**Author Contributions:** Conceptualization, W.N.R.; Data curation, W.N.R.; Formal analysis, R.A.R. and W.N.R.; Funding acquisition, W.N.R.; Investigation, N.N.T.S., R.A.R., R.A., H.A., T.T., M.N. and H.M.; Methodology, H.A., R.S. and W.N.R.; Project administration, N.N.T.S. and R.A.R.; Resources, K.A.R., M.G., R.S., T.T. and W.N.R.; Supervision, W.N.R.; Validation, M.G. and W.N.R.; Visualization, N.N.T.S. and R.A.R.; Writing—Original draft, W.N.R.; Writing—Review and editing, N.N.T.S. and W.N.R. All authors have read and agreed to the published version of the manuscript.

**Funding:** This work was funded by Ministry of Higher Education Malaysia (FRGS/1/2020/STG07/ USM/02/2).

**Institutional Review Board Statement:** Not applicable.

**Informed Consent Statement:** Not applicable.

**Data Availability Statement:** Data available for this study are presented in this article.

**Acknowledgments:** The authors wish to thank the staff of Nuclear Medicine, Radiotherapy and Oncology Department, Hospital USM and Hyogo Ion Beam Medical Center for helping in conducting irradiations for this experiment.

**Conflicts of Interest:** The authors declare no conflict of interest.

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
