# Peer review of "Gafchromic™ EBT3 Film Measurements of Dose Enhancement Effects by Metallic Nanoparticles for 192Ir Brachytherapy, Proton, Photon and Electron Radiotherapy"

_radiation, doi:10.3390/radiation2010010_

Round 1
Reviewer 1 Report
The authors have addressed a very interesting problem on cancer treatment. They have explored the action of different nanoparticles on the control of dose during radiation treatment of cancer. The article is well written and conclusions are well supported by the observations. Hence, may be considered for publication. Authors need to clarify the following points before publication:
1- There is confusion between Bi and Bi2O3. The authors need to rectify the typo error in the manuscript.
2- What is the core and size in table 1? What is the error on average particle size? Micrographs on particle size may enhance the description of the article and future reference by others. The parameters are not described in the caption of the tables.
3- A better photograph of GafchromicTM EBT3 Films may be given. May be separated into two or three figures with a bigger size.
Finally, I recommend considering the article for publication.
Author Response
Thank you very much for the reviews.
Please see the attachment, for the responses and revision.

Reviewer 2 Report
This presented word addressed a good idea by mixing nanotechnology and radiotherapy field.
The idea seams relevant to the field as it could enhance the dose effect of different radiotherapeutic techniques.
It is better for the author to clarify why the tested only metallic nanoparticle and not consider other types of nanoparticles and why they had chosen iron, gold and bismuth NPs.
Is it possible to test this idea on animal model?
Reference section seems good and relevant to the topic.
Figure 5: it is not clear
Figure 6: explain why there is no any relevant difference between testing different energetic radiotherapeutic techniques for the same NP.
Author Response

(The authors gave the same response as above.)

Reviewer 3 Report
The authors present an extensive and interesting work on the quantification of physical DE using GafchromicTM EBT3 RCFs with three different types of NPs.
I have found minor typos and have few suggestion:
- figure 4 - please use different colours or other ways to better display the curves
- fgure 4 - add error bars or indicate in the text the accuracy of these measurements in the text
- table 5 - add fitting equations
- figure 5 - add a better way to display the curves
- table 6 - the numbers cannot have 3 digits after the comma if the standard deviation has only two, please correct.
Author Response

(The authors gave the same response as above.)

Reviewer 4 Report
Dear Authors,
I think you presented a good manuscript. There are some minor English and editing problems, like often the brackets are missing from the references, the significant digits in Table 6 and some sentences are a bit off or self-contradictory in my opinion.
I have some questions as well, the major one is on the assumed water - D-PBS equivalency, please provide some reference or experimental evidence that you can do that, I am not certain, but the amount of Na it has in it might cause some slight effect for photons from Ir-192 or with even less energy, however this might be negligible for your purposes.
Also I suggest to repeat some of your more important conclusions/statements from the discussion section in the conclusion part, like "The use of EBT3 RCFs may provide an adequate detection geometry for highly energetic PIXE/PIGE, but most definitely not suitable for the weaker and shorter-ranged Auger electrons due to the thick polyester substrate." or that despite the lack of significant physical DEF, the NPs have an effect, because the biological effects are mainly ROS based and NPs generate more of those, because I think these are more important and give more context than writing the equivalent of we measured a lot of things, which is not really a statisfactory conclusion.
Author Response

(The authors gave the same response as above.)
